# Identification of Problems Associated with the Usage of Friction (Koeppe) Hoists Based on Geodetic Measurements

**Agnieszka Ochałek [1],\*, Wojciech Jaśkowski [1]**  **and Mateusz Jabłoński [2]**

[1]   Faculty of Mining Surveying and Environmental Engineering, AGH University of Science and Technology, Adama Mickiewicza 30, 30-059 Kraków, Poland; jaskow@agh.edu.pl

[2]   AMC ZRGiW, Mników 389, 32-084 Morawica, Poland; mjablonski@amc.net.pl

\*   Correspondence: ochalek@agh.edu.pl

**Abstract:** The hoist assembly based on the Koeppe friction is a commonly used solution in mining. However, it has some disadvantages. A few centimeters offset of the groove axis can lead to excessive abrasion of linings on the Koeppe friction and pulleys. As a consequence, the mines are forced to bear the direct and indirect costs of replacing the linings such as the cost of materials and service as well as the cost of extended machine and shaft downtime. Last year, the authors undertook a geodetic inventory of the condition of two hoisting machines with a Koeppe winder. Terrestrial laser scanning enhanced with precision total station measurements were performed. Additionally, elements particularly important for the performed analysis (inclination of hoisting machine and rope wheels shafts) were determined by the precision leveling technique. Obtained results were verified using measurements on Szpetkowski's tribrach. Appropriate selection of the measurement methods in both analyzed examples allowed us to determine the causes of destruction of each hoist assembly component. Based on precise geodetic data, guidelines have been defined for rectification (twisting and shifting the rope pulleys), which seems unavoidable despite the lack of unambiguous legal regulations.

**Keywords:** Koeppe; mining surveying; 3D laser scanning; GNSS; shaft hoist assembly; geodetic inventory; headframe; point cloud processing; TLS

## 1. Introduction

The shaft hoist assembly, which consists of the hoisting machine, the headgear, and the shaft, is the core of underground mine manufacturing. Exceeding the value of tower inclination, rope friction angles, or excessive inclination of the pulley shafts and hoisting machine may lead to critical failures [1]. The measuring service periodically or continuously monitors the condition of the hoist assembly geometry, indicating the rectification process. Many studies related to different aspects of mine hoist and friction safety [2–8] have been conducted at mines In this article, the authors focused on a comprehensive geodetic inventory with the determination of the measuring methodology of the hoisting device with a propeller (the Koeppe system) and two levels of rope pulleys associated with it (upper and lower) [9]. The process of conducting geodetic control measurements on the example of two hoisting assemblies with the Koeppe system is presented.

The geodetic control methods used so far have been usually based on various plumbing methods. Few automation trials have allowed for the determination of one or two elements of the geometric control of the shaft hoist assembly. However, to make a full inventory of the mining shaft, several complementary classical measurement techniques have to be carried out. As a consequence, such an

approach significantly extended the performance of the surveys. Currently, the set of classical geodetic methods can be replaced with laser scanning, which allows obtaining as much necessary information as possible in the shortest possible time [1,10]. This approach is visible in every branch of industry where it is necessary to obtain as much precise data as possible in the shortest possible time. Laser scanning could be used in testing the clearance gauge of railway [11], maritime measurements [12], monitoring of roads and bridges [13] or building [14] state, concrete structures [15], and in many other industries. Naturally, this technology has also been developing in the mining industry for over 20 years [16,17]. The scope of the data obtained with this method is incomparably greater than that obtained with traditional methods. Furthermore, the use of scanning may be important from an economic point of view as well as for the safety of employees [18]. The inventory measurements of the hoisting machine and rope wheels, along with the determination of the tower and shaft inclination, can be performed by a four-person measuring team even during one shift. To have a comprehensive inventory, only the precise leveling of the shaft's cable pulleys and winding machine, together with establishing the geodetic network to link the model to the local or global system, is required to be performed with the help of classical methods.

## 2. Materials and Methods

In December 2019, several hoisting assemblies were inventoried by the surveying team consisting of AMC company (Leawood, KS, USA) and AGH University of Science and Technology representatives with extensive experience in this type of research (e.g., [1]). Two of the measured objects had problems associated with the usage of Koeppe friction. The measurements were performed using a panoramic laser scan with the phase scanner FARO FOCUS 3D. The device allows information to be obtained about the measured object with a resolution of 5 mm/10 m distance with an accuracy not worse than ±2 mm for a single measuring station. Additionally, elements particularly important for the performed analysis (inclination of hoisting machine and rope wheels shafts) have been determined by the precision leveling technique (with the usage of Trimble DiNi 0.3 and the set of invar rods that ensures leveling accuracy below 0.3 mm/1 km). The internal consistency of the point cloud (Figure 1) was ensured by the high accuracy measurements of the geodetic network with a total station (Trimble C3 with an accuracy of 2″ angle and length of 2 mm + 2 ppm.).

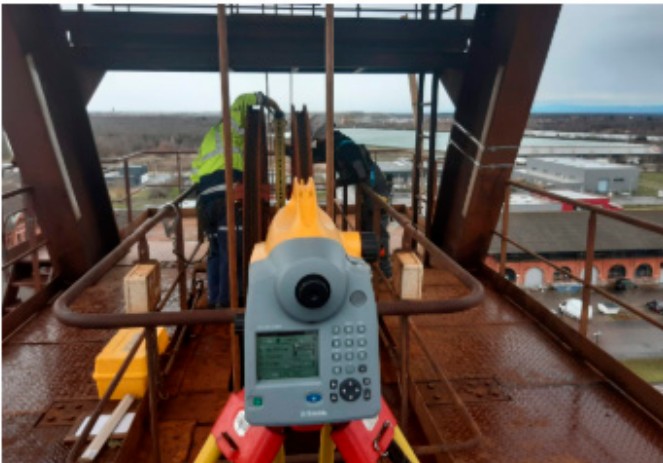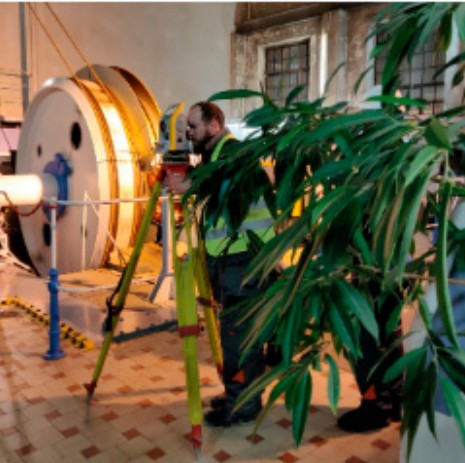

**Figure 1.** Measurements carried out at the headframe (on the left) and hoisting machine (on the right).

Another important issue was the problem of uneven lining wear and a high probability of excessive tower vibrations. Therefore, the dynamic tests of headframe inclination during its operation and emergency breaking was performed. Simultaneously, the rope movement amplitude was also tested. The test of the headframe inclination was carried out with the use of two measurement technologies:

laser scanning and GNSS measurement in real-time kinematic mode with a receiver located on the shaft tower (Figure 2) with the base station recording data in the static mode [19].

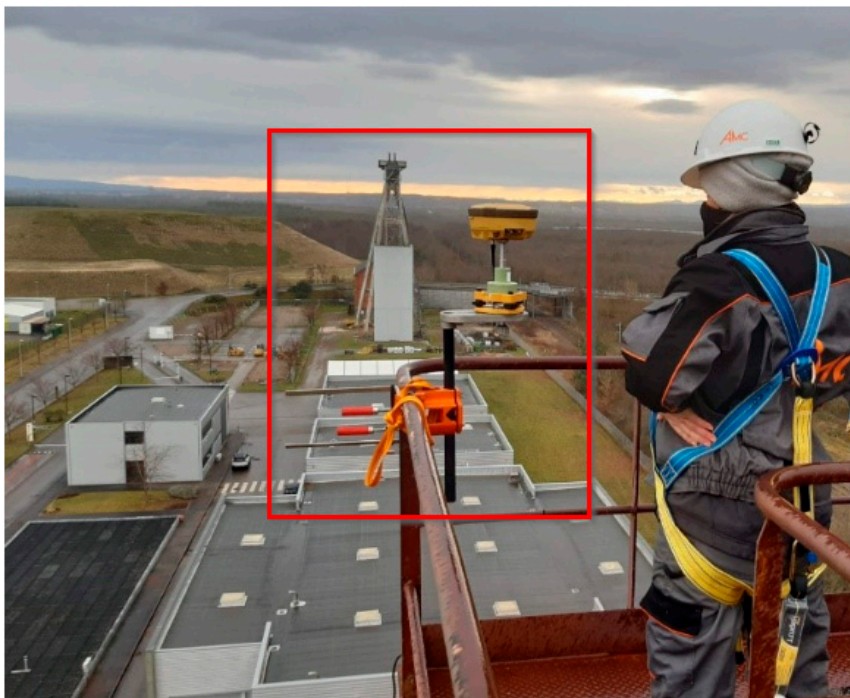

**Figure 2.** Receiver measuring vibrations of the headframe in the real-time kinematic (RTK) technique.

The results of the laser scanning measurements (point clouds) allowed us to develop spatial models of inventoried objects [20] (Figure 3). In addition, subsequent computer-based analyses were performed such as determining the characteristic points located on the axles of hoisting machines and cable pulleys, establishing transverse profiles on shaft towers, and the amplitude of vibration of the winding rope. The obtained base of points was used to prepare the spatial arrangement of the axes of the Koeppe drums, pulleys, shaft tower structure, and ropes.

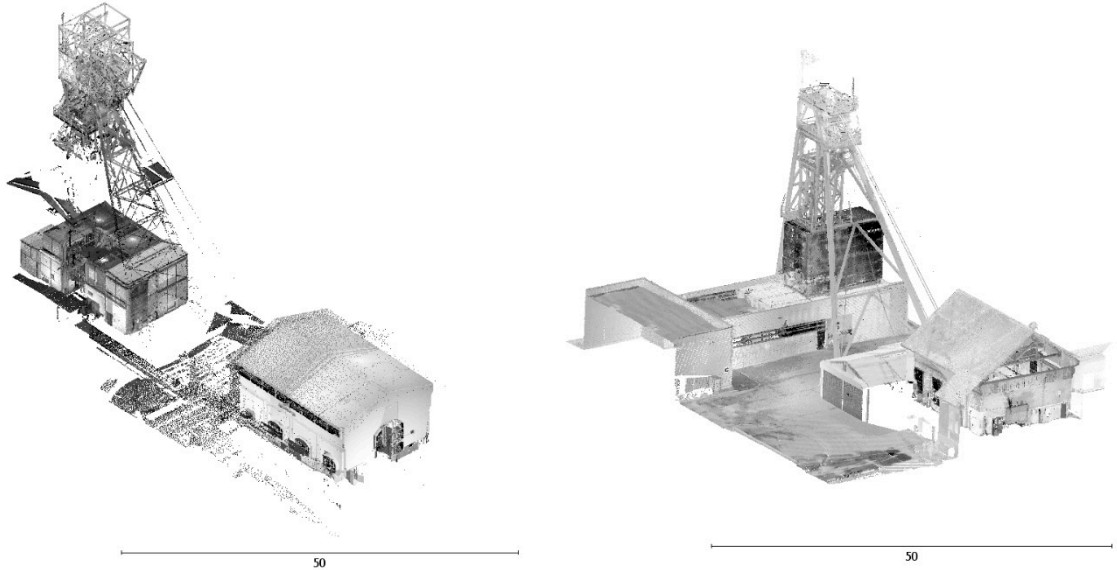

**Figure 3.** Point clouds of the inventoried headframes (located in Poland on the left, located in France on the right).

All the obtained data were finally used to determine the friction angles of the ropes on the winding wheels, the inclination of the pulley shafts and the winding wheels, determine the verticality of the shaft tower (including during its operation period and emergency breaking), and determine the verticality of ropes (descending from the level of cable pulleys to the shaft room). All together, these data allowed us to determine whether the inventoried objects could still be safely used according to the law. Moreover, they constitute a database necessary to perform the rectification of individual elements of the hoist assembly and form an excellent source needed to complete the engineering documentation of the mine.

## 3. Applicable Law Regulations in Relation to the Inventoried Objects

Every kind of geodetic survey should be carried out with relation to the applicable law (applicable in the country where the inventoried object is located). Therefore, it is crucial to be aware of the applicable law regulations in a given topic. The calculation of apparent angles of rope friction (measured on a horizontal plane) is associated with calculating the differences of the azimuths (directional angles) of the axes characterizing the winding device. In applicable Polish law, the only reference to the case of hoisting assembles is present in the notation from the Regulation of the Minister of Energy on detailed requirements for the operation of underground mining plants (Annex 4, point 3.11.17): "In the rope wheels on the tower shaft hoists with a winder or a bobbin hoisting machine, the symmetry plane of the pulley groove coincides with the plane defined by the axes of oncoming and converging rope" [21].

In the case of a hoisting device with a Koeppe friction and two rope pulleys (upper and lower) associated with it, the friction angles (and shifts of the proper grooves) are determined separately for the upper rope and the lower rope. Therefore, for the upper rope on the upper wheel, the friction angle is determined as the difference in the direction angles of the longitudinal axis of the pulley and the individual pulling axis associated with that pulley. For this upper rope, the friction angle on the Koeppe winder is determined as the difference in the direction angles of the longitudinal axis (plane) of the Koeppe winder and the pull axis associated with that wheel. The apparent friction angles for the bottom rope are determined analogously. In a situation where the hoisting device is a multi-rope friction winder, the apparent rope friction angles are determined for each pair of corresponding grooves on the drive wheel and cable wheel. Two friction angles are determined for each pair, one for the rope on the drive wheel and one for the rope on the cable wheel. To determine these friction angles, it is necessary to know the azimuths (direction angles) of the pull axis for the corresponding set of grooves on the drive wheel and on the cable wheel. This pull axis is defined as the straight line connecting the center of the rope groove on the drive wheel with the center of the pulley groove. The friction angles of the rope on the driving wheel are therefore the differences between the azimuths of the individual grooves on this wheel and the azimuths of the pull axis associated with each of these grooves. Similarly, rope friction angles on cable wheels are the differences between the azimuths of the grooves on those wheels and the azimuth of the pull axis associated with the individual wheel (Figure 4).

In the case of the Koeppe winder and two rope pulleys connected to it (positioned next to each other), the angles of friction and the displacement of the grooves are determined separately for the underlap rope and the overlap rope. The friction angle is determined as the difference in the direction angles of the longitudinal axis of the pulley and the rope associated with that pulley. For this rope, the friction angle on the Koeppe winder is determined as the difference of the direction angles of the longitudinal axis (plane) of the Koeppe winder and the pull axis. However, it is not possible to comply with the provision contained in the regulation cited above. Therefore, it should be assumed that the drawing axis should be symmetrical (in the middle between) to the planes of the pulleys (Figure 5).

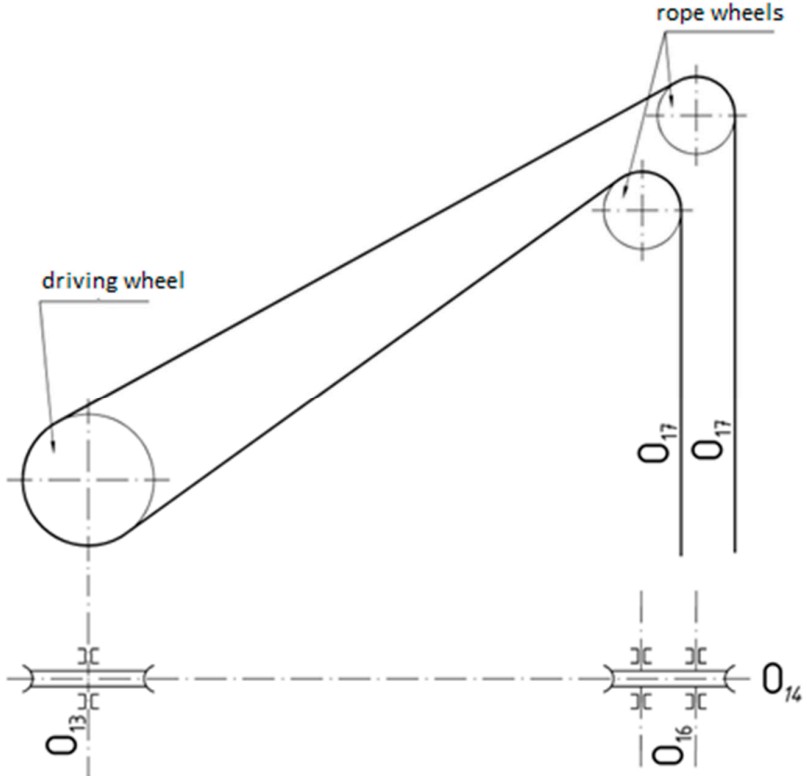

**Figure 4.** Sketch of the hoisting axle for the machine with the drive wheel (Koeppe) of the pulleys located in one vertical plane.

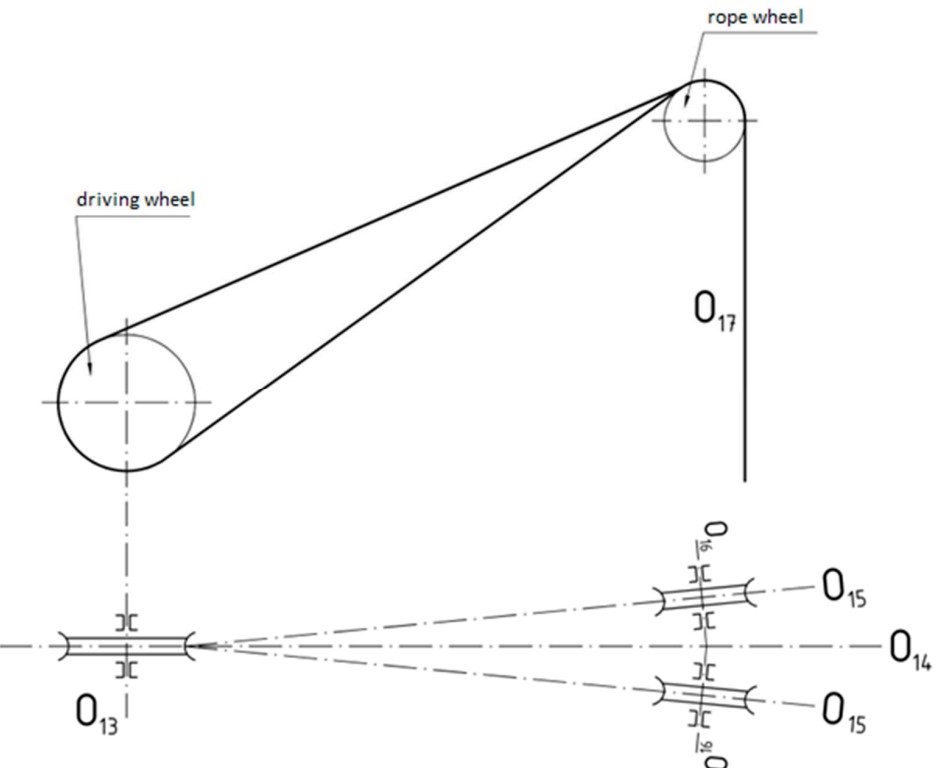

**Figure 5.** Sketch of the hoisting axle for a machine with a drive wheel (Koeppe) and pulleys arranged side by side.

In all cases, two axes—the shaft axis of the winding machine ($O_{13}$) and the drawing axis ($O_{14}$)—are significant to determine the geometrical conditions associated with the winding machine and driving wheels on the shaft tower (Figure 6) and are defined as follows:

- $O_{13}$—axis of the driving wheel shaft—longitudinal axis of the driving wheel shaft;
- $O_{14}$—draw axis—straight line which in the case of pulleys positioned in one vertical plane connects point of intersection of the transverse groove plane with its shaft axis ($O_{13}$) with the point of intersection of the pulley surfaces with their shaft axes ($O_{16}$); and
- $O_{17}$—hoisting machine suspension axis—vertical line passing through the center of the rope cross-section at the point where the rope exits the rope pulley.

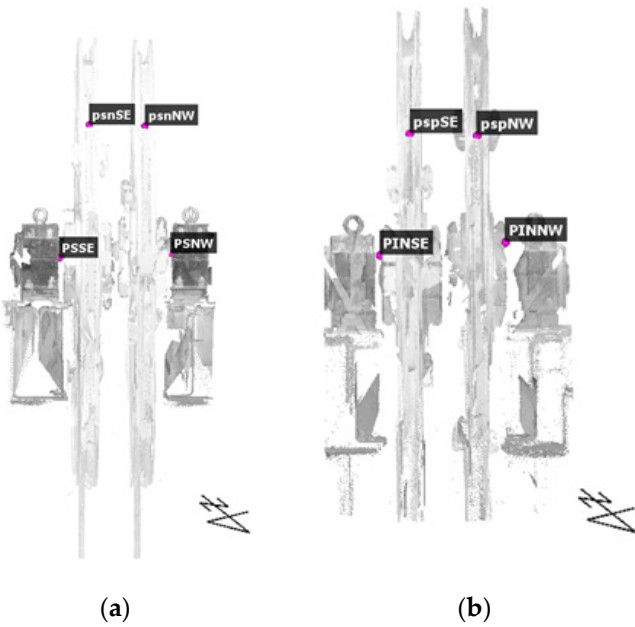

(**a**)　　　　　　　　　　　　　　　　(**b**)

**Figure 6.** Location of the rope winding points on rope wheels, platform: (**a**) upper; (**b**) lower.

The first inventoried object was an A-form double backstay headframe with a multi-rope friction Koeppe winder and two levels of wheels (Figures 6 and 7). Multi-rope friction winders are tower mounted, with either cages or skips, and provided with a counterweight. Four ropes are used, operating in parallel and sharing the total suspended load. The hoisting assembly is located in the east of France and supports the closed potassium salt mine. Although currently the shaft tower and the shaft itself are not heavily used, in the near future, a significantly higher load of the tower is planned, both in terms of travel frequency and weight of transported materials. The mine's power engineering department has diagnosed the problem of uneven lining wear at the upper level of the pulleys. According to the observations, at a higher winding speed of the cage (the maximum allowed speed in this analyzed problem was 4 m/s) and a larger load mass, the overlap rope exceeded normal movement amplitude, and the tower had been vibrating more than regularly. This second aspect should not affect the operation of the tower with a steel structure (especially with the low height of the lower and upper pulleys above the shaft station—24 and 28 m). However, it was also decided to analyze this issue. It is worth mentioning that French law does not raise the issue of using shaft towers, periodic technical inspections, or of critical values of deviations. It is the manager's responsibility to maintain the technical condition of the facilities in an appropriate condition. The mine administrator decided to base the assessment on the Polish regulations and experience of the measuring team.

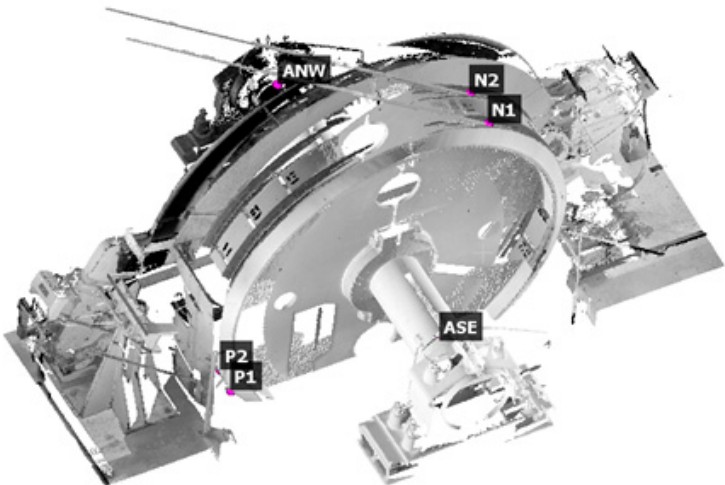

**Figure 7.** Location of the characteristic points on the Koeppe winding machine drum.

The second analyzed object was a steel structure shaft tower with a single-rope friction Koeppe winder and pulleys arranged side by side (Figure 8). The shaft tower is situated in Poland (Lower Silesia), therefore, the authors performed their surveys and analyses according to Polish law. From the beginning of the mine's operation, there were two shafts spaced 20 m apart, with two symmetrical perpendicular two-post head frames connected with one headroom building above these shafts. As a result of the mine reconstruction, one shaft and its shaft tower were removed. As a consequence, the attached headroom building was demolished. One compartment was left in the shaft (skip had been removed) and as a result, the perpendicular strut had been dismantled. Now, the shaft tower that was measured looks like one-post head frame and its construction is significantly weakened. The consequences of the above changes can be observed in the inclination of the construction.

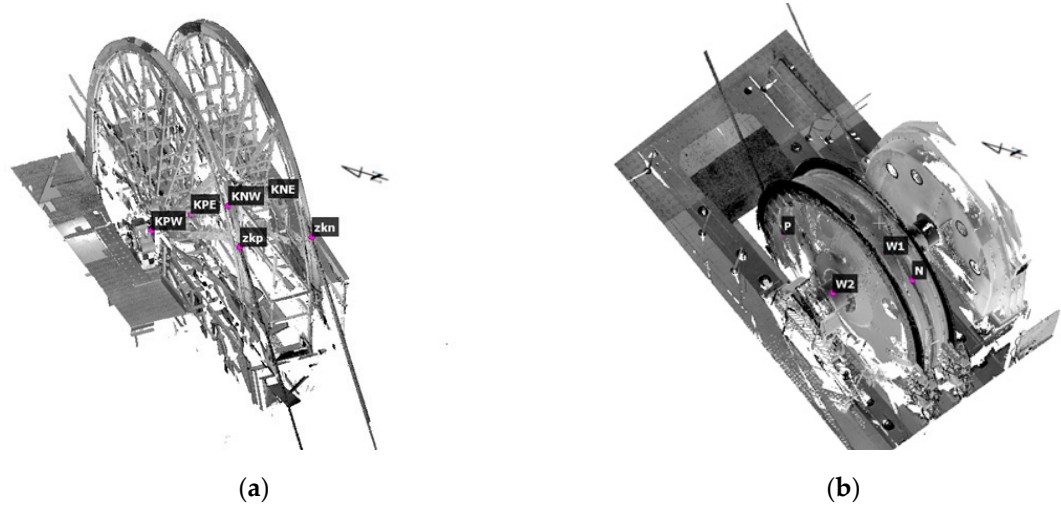

|           (a)           |           (b)           |

**Figure 8.** (**a**) One-rope winding machine, (**b**) two pulleys located next to each other on one platform.

Due to the occurring and diagnosed problem (lining abrasion during the operation), both cases in France and Poland are similar. In simplified terms, it can be assumed that the corresponding shaft axles on the propeller plate and the cable pulleys are not collinear. The issue has been temporarily solved by dredging the groove, the place where the rope is located during the exact measurement/inspection. However, after a while, the rope rolls further and creates a new groove. As a consequence, oversize clearances and asymmetrical tensions are created in the grooves on the drive wheel and in the pulleys, which results in a different distribution of forces than were originally planned. This is an extremely dangerous situation, because the uneven loading of the drive wheel and rope wheels can not only affect

lining wear, but also the structural unforeseen loading of the head and shaft tower as well as abnormal vibrations. In the worst case scenario of buckling of the shaft and displacement of the descent points of the ropes to the shaft, this in turn can lead to asymmetrical guiding of the shaft cage and critical failures related to it.

## 4. Results

To diagnose the problem and show the value of the groove offsets, very precise geometry analysis is crucial. It is necessary to identify not only the points of departure of the ropes from the propeller shaft and the pulleys, but also the twisting of these elements related to each other and to their shafts. Thus, on very short bases, drawing azimuths are even extrapolated to several dozen meters. This process requires measuring many more points, not just characteristic ones, as it is done using the total station measurements, or the earlier used "ordinate and cut" method. This approach clearly indicates the usage of laser scanning, which allows a point cloud to be generated, representing the entire object and its surroundings in a high resolution. Then, based on the modeling of the point cloud, a spatial model of the shaft axis, wheel surfaces (Figure 9a), and propeller shaft, ropes, and the tower itself with its core (Figure 9b) is created.

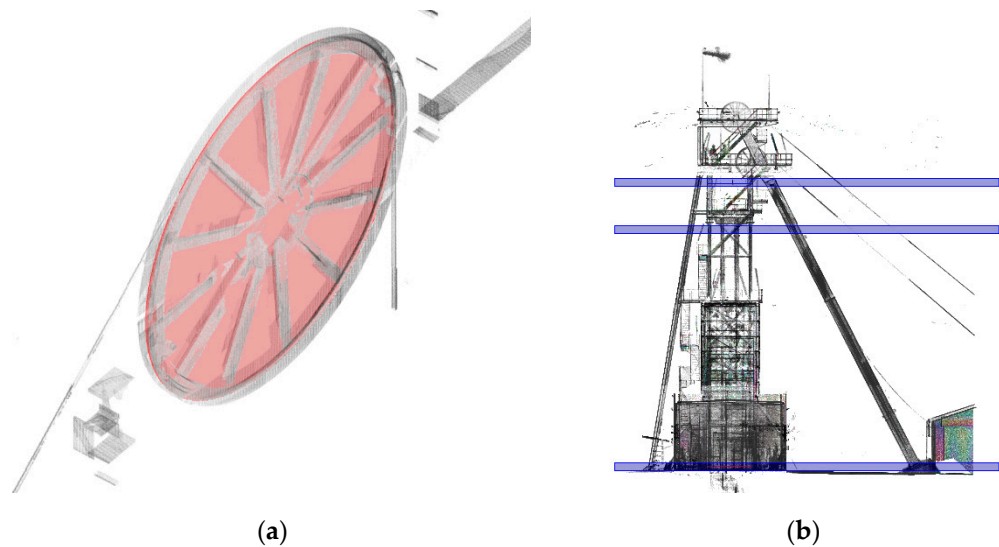

(**a**)    (**b**)

**Figure 9.** Point cloud processing: (**a**) modeled plane of the pulley groove, (**b**) determination of cross-sectional levels for testing the inclination of the tower and its core.

As mentioned earlier, most of the elements crucial for the analysis were determined twice using different measuring methods (Table 1). An error in determining a single point was adopted to carry out an accuracy analysis in order to compare the used measurement methods. However, while relating the total station and laser scanning methods, it must be said that determining a point with a total station does not guarantee the perfect defining of the object. It is not possible to perfectly pick out the axis of the rope or the groove in the field, but it is possible to do this on a spatial model (point cloud) of a given object. Moreover, the measurement time itself was analyzed and thus the time of excluding the object from motion. This is a very important aspect in the context of shaft facilities, the closure of which generates logistical problems and costs. The last important queries were to compare the economic aspect of processing the data from every method and specifying the information that each method provides for individual analyses. Table 1 summarizes the averaged errors in determining the coordinates of points or azimuths of pairs of points used for the analysis of subsequent objects, at the same time marking the data sources used for the analysis.

**Table 1.** Averaged errors in determining the coordinates of points or azimuths of pairs of points used for the analysis of subsequent objects (at the same time marking the sources of data taken for the analysis).

| Construction Element/Measuring Method | Terrestrial Laser Scanning | Tacheometry | Precise Levelling | Szpetkowski's Tribrach | GNSS |
|---|---|---|---|---|---|
| tower verticality | 6.0 mm | - | - | - | - |
| determination of the groove axis offset | 7.4 mm | 6.5 mm | - | 3.1 mm | - |
| shaft tower vibration amplitude | 4.0 mm | - | - | - | 2.0 mm |
| applies to both drive pulley and rope wheels: | | | | | |
| rope descent from rope wheels points | 5.8 mm | 5.2 mm | - | - | - |
| rope descent from friction disc points | 4.7 mm | 3.9 mm | - | - | - |
| rope wheels' grooves azimuths | 107 mgon | 96 mgon | - | - | - |
| friction disc's grooves azimuths | 69 mgon | 62 mgon | - | - | - |
| ropes azimuths | 78 mgon | 70 mgon | - | - | - |
| shaft inclination | 2.7 mm | - | 0.3 mm | - | - |
| determination of the shafts base | 2.7 mm | 2.2 mm | - | - | - |

The basic analysis of the table in terms of the methods used for obtaining the information clearly indicates that the most universal measurement technology is terrestrial laser scanning. Properly conducted post-processing of the point cloud enables all the data necessary for the mining documentation to be provided, with the reservation of a lower accuracy of determining the inclination of the shafts of the hoisting machine and cable pulleys. Therefore, it is necessary to supplement this part of the survey process with precision leveling. The precise leveling itself is not very time-consuming: depending on the shafts' accessibility, it should take up to 15 min for each element (for the machine shaft and each level of the pulleys).

The technologies with the highest coverage in the above table were the laser scanning and total station measurements. They allowed us to obtain information about most of the elements required in the inventory process. Nevertheless, the method used up to now to determine most of the elements (tacheometric) has been creating many issues. Elements such as the axis of the rope, the groove, or the plane of the drive disc are represented only by several points. As a result, a created point cloud contains a small number of supernumerary observations. Individual axes are fitted using averaging methods into these thin point clouds. The laser scanning gives a similar accuracy of measurement, but each element is represented by millions of points, so you can use methods of suppressing outliers and modeling the element with analytical methods.

In fact, only the parts that should be analyzed are measured with a laser scanner. The access to the shaft pit, the levels of the rope pulleys, or the hoisting machine building is often very limited and combining the scans by methods of fitting them into spheres, planes, or characteristic points (and even a cloud to cloud register) would be too inaccurate. In addition, it is usually required to precisely reference measurements, and thus to determine characteristic elements such as rope azimuths, drawing azimuths, azimuths of shaft axes, azimuths of the planes of rope pulleys, and the driving drum in a global or local (mine) system. Therefore, it is necessary to connect successive scanner stations or groups of scanner stations (only and independently are measured: hoisting machine, rope pulley levels, and shaft at the shaft room level). This is done by measuring the geodetic network (using the multi-tripod method to minimize the impact of re-centering) and—already in reflectorless mode—the black and white reference targets. In flat terrain such as the transition from a shaft yard to a shaft room or to a hoisting machine building, the angle measuring errors are much smaller than in steep target measurements carried out when measuring the warp or determining the discs at the level of the rope pulleys. The error in determining the characteristic points using the tacheometric method is therefore the same as the error generated by the exact alignment algorithm. From the propagation of errors, it can be assumed that the error in determining the coordinates of points by laser scanning can be equal to the square root of the error in determining the point within one group of scans (point clouds) and the error in determining the reference targets that fit these scans to the model [22].

The same applies to determining the azimuths of ropes and the grooves of the drive disc and rope pulleys. The accuracy of determining the azimuths after the geodetic network is elevated to the levels of the pulleys is lower, and the errors determined by the laser scanning method are correspondingly greater according to the propagation of errors.

The inclination of the headframe can be carried out independently by making a measurement with a laser scanner around the tower, or referring it to the local or global system using geodetic methods. With a correctly performed measurement and averaging and making accurate sections, the value of the inclination (and its path) of the headframe axis was 6 mm.

## 5. Discussion

In the first analyzed case (France), the upper rope wheels required rectification, but the entire object was inventoried. Thanks to the geodetic points network situated at the shaft room and its surroundings, it was possible to determine the coordinates of black and white scanning targets. These targets were situated at each level of the pulleys, shaft station, and the building with the winder drum and were used to register the point cloud and georeferencing. In addition, the coordinates of characteristic points at the tower shaft construction were determined with reflectorless measurement for better control of the results. Obtained point clouds were post processed. Azimuths of the planes and grooves of the pulleys and the propeller shaft, and also rope and shafts axes were obtained using the spatial model located in the local coordinate system. The azimuth values allowed the determination of rope friction angles on the pulleys and the winding pulley. A digital 3D model of the object was made, which enables a determination of the axis shifts of the grooves. The lower wheels converged with the drive wheel disc at 13 mm, and the upper wheels turned out to be twisted relative to the propeller. Ropes coming out of the winding pulley grooves perfectly hit the upper rope wheels, but when twisted, they showed non-parallelism. The extension of the axis of the grooves of the upper pulleys for the winding pulley showed an error of 92 and 128 mm. Therefore, for rectification, it is required to slightly twist and move the arrangement of the upper pulleys while maintaining the position of the current points of departure of the rope on the winding machine. Changes in the points of descent of the rope to the shaft indicates that its deflection from the vertical will improve.

The second case analyzed (in Silesia) comes down to the verification of the condition that the draw axis should be symmetrically placed in the middle between the rope wheel planes properly twisted to the machine. After analyzing previous surveys and discussions with the mine's power engineering department and surveying-geological department, it turned out that in the context of the displacement of the grooves relative to each other, the deflection of the tower does not play a role because it shows a vector directed toward the winding machine. The research showed that the deflection of the object was within the scope of the permit (1/500 tower height in accordance with the relevant Regulation [21]). The measured displacement of the groove axis on the drive wheel related to the draw axis at the wheel level was 166 mm, whereas the displacement of the groove axis on the drive wheel in relation to the pull axis at the level of the propeller wheel was 425 mm for the underlap rope's wheel and 47 mm for the overlap rope's wheel (Figure 10). The geometry of the object also indicates that the inclination values of ropes going down to the shaft may be problematic in the future. Therefore, the values indicate that the shaft tower should be rectified first. Only after that, should the whole hoist assembly be re-measured and the pulley assembly rectified adequate to the results of the inventory.

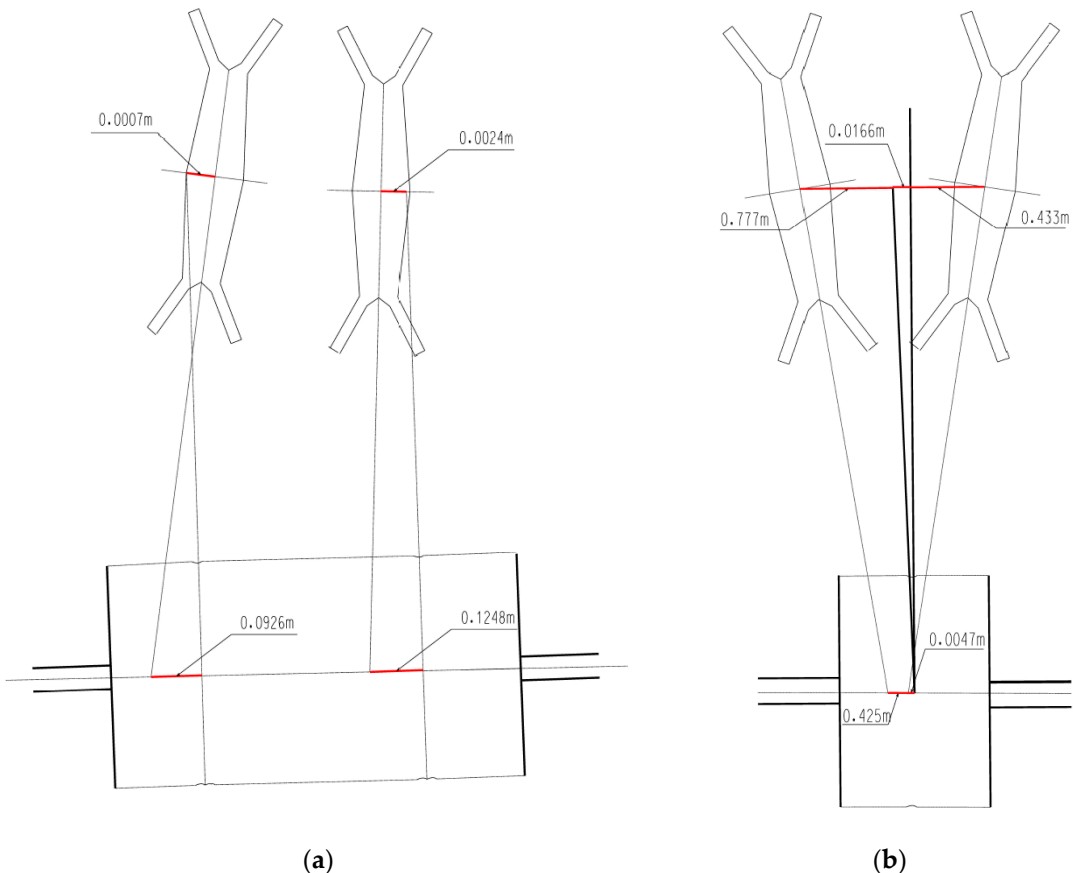

**Figure 10.** A sketch of the axis shifts of the corresponding grooves at: (**a**) upper pulleys relative to the winding machine in France, and (**b**) cable pulleys relative to the winding machine in Silesia.

## 6. Conclusions

The Polish law [21] imposes appropriate deadlines for inventorying the geometry of the hoist assembly. Every five years, the inclination of the shaft tower, geometrical relationships between the pulleys, and the winding machine as well as the straightness of the guides and the shaft geometry should be determined. In the opinion of the team of authors, such a study should be combined in the scope of checking the inclination of the shaft tower together with the shaft and winding machine due to the possibility of conducting a wider, holistic analysis. In addition, it is worth mentioning that many mines are deciding to measure the tower inclination more often, even every year.

Thanks to the technological progress in the field of hardware and software, the appropriate selection of measurement methods allows faster and more accurate inventories. The presented examples propose a modern approach to the documentation of the mine object process. Moreover, the collected data can be later used for renovation, extension, or decommissioning design work. The context of a hoist assembly with friction Koeppe winder is particularly demanding. A several centimeters axes offset can affect the geometry and fluidity of the entire construction. As mentioned in this article, the costs of the improper operation of such a system can be very high, and the usage tedious. It is therefore particularly important to systematically control geometrical relationships and conduct possible rectification.

The limitations resulting from the accuracy of the measurement with a laser scanner and combining subsequent scans indicate the need to use methods such as precision leveling and tacheometric measurement of the matrix and connection points. As shown in the article, the errors from the combination of the above methods are several orders lower than the limit values for the use of objects.

**Author Contributions:** Conceptualization, W.J. and M.J.; Data curation, A.O.; Formal analysis, W.J. and M.J.; Investigation, A.O. and M.J.; Methodology, W.J.; Project administration, A.O. and M.J.; Supervision, W.J.; Validation, W.J.; Visualization, A.O.; Writing–original draft, M.J.; Writing–review & editing, A.O. All authors have read and agreed to the published version of the manuscript.

**Funding:** This research received no external funding.

**Conflicts of Interest:** The authors declare no conflict of interest.

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
