# Peer review of "Identification of Problems Associated with the Usage of Friction (Koeppe) Hoists Based on Geodetic Measurements"

_geosciences, doi:10.3390/geosciences10110444_

Round 1

Reviewer 1 Report

The authors presented a project of geodetic surveys using a laser scanner. The method and the procedure are well discussed. I have a few questions and comments for the authors. After they are addressed, I would recommend that this manuscript can be accepted by geosciences.

1. The captions of the figures are not clear. For example, figure 1 is missing explain for the right and the left. Figure 3 and 9 are missing a scale bar. What is the difference between the left and right in figure 3? In figure 2, it would be better for highlight the sensor.

2. The laser scanner can only measure one point at a time. How long did the whole survey cost?

3. I don’t think the introductions gives a clear motivation of this research. I recommend the authors should explain more clearly.

Author Response

1. The captions of the figures are not clear. For example, figure 1 is missing explain for the right and the left. Figure 3 and 9 are missing a scale bar. What is the difference between the left and right in figure 3? In figure 2, it would be better for highlight the sensor. - Corrected.

2. The laser scanner can only measure one point at a time. How long did the whole survey cost?

The scanner used in this research (FARO Focus 3D) measured up to 1 million points per second. Measurements on one laser position took about 3-4 minutes (it depends on the chosen resolution of the scan). To make surveys on all levels (shaftroom, headframe - upper and lower rope wheels and around the hoisting machine) it took around one mine shift. 

3. I don’t think the introductions gives a clear motivation of this research. I recommend the authors should explain more clearly. - the authors have expanded the introduction with additional information.

Reviewer 2 Report

Dear authors

In the text, the authors made a geodetic inventory of two hoisting machines from Koepe winder. A terrestrial laser scanner and a total station were used for the measurements. In my opinion, the text is in the form of a measurement report and adds little to science. However, the text may be interesting for the reader after extending and undertaking scientific dissection with the authors of previously published works.

L27: Keywords: Too few keywords. The keywords "mining surveying", "3D laser scanning" are too general.
L29: Introduction: The introduction lacks an overview of the state of science in the described scope. The examples given in the text do not differ from laser scanning of engineering structures (bridges, water lock gates, GSM relay towers), so it is worth looking for examples and improving the text by analyzing already published works, e.g. in Geosciences, Sensors or Remote Sensing
L152, L155: Please correct the description in the picture. Description in English is required.
L235: Please use a period instead of a comma to separate the decimal part of numbers.
L350: References. The review is very modest, the cited publications are very old. Some of the items are not available to the reader (e.g. Polish). The translatable phrases must be translated into English. Under item 13, a more detailed reference should be made to the official journal of laws.
I recommend extending the literature review, referring to items from other Polish and foreign research centers. It is worth reviewing the texts of Polish authors (eg Tysiac P., Janowski A., Mikrut S., Osinska-Skotak K.) and numerous foreign authors.
Please look at the MDPI template and correct the presentation of the references (https://www.mdpi.com/authors).

Author Response

Dear Reviewer,

All the points from your list were detailly checked and expanded, revised or changed and as follows:

L27: added new Keywords: Koeppe, mining surveying, 3D laser scanning, GNSS, shaft hoist assembly, geodetic inventory, headframe, point cloud processing, TLS

L29: expanded as you recommended by examples from Geosciences, Sensors and Remote Sensing. Positions in the references: 2-17.

L152, L155: description changed to English.

L235: period instead commas.

L350: References expanded and corrected.

Round 2

Reviewer 1 Report

The authors has addressed my questions and comments. I recommend that this manuscript can be accepted by GeoSciences.

Author Response

Thank you very much.